# Asynchronous tele-expertise (ASTE) for prenatal diagnosis is feasible and cost saving: Results of a French case study

M'hamed Beldjerd[1], Antoine Lafouge[2], Roch Giorgi[3], Anne-Gaëlle Le Corroller-Soriano[1]*, Edwin Quarello[4,5]

1 Inserm, Aix Marseille Université, IRD, SESSTIM, ISSPAM, Marseille, France, 2 Cabinet de Gynécologie et Obstétrique Hyères, Hyères, France, 3 Aix Marseille Université, APHM, INSERM, IRD, SESSTIM, ISSPAM, Hop Timone, BioSTIC, Biostatistique et Technologies de l'Information et de la Communication, Marseille, France, 4 Centre Image 2, Marseille, France, 5 Service de Gynécologie Obstétrique, Hôpital Saint Joseph, Marseille, France

* anne-gaelle.le-corroller@inserm.fr

**Data Availability Statement:** Our Data are available and have been added as Supplementary files.

## Abstract

### Objective

The objective of this study was to assess the potential of the use of asynchronous tele-expertise (ASTE) to provide prenatal diagnosis from a medical and economic point of view.

### Population

Patients screened by a midwife at a primary center.

### Methods

A technical and clinical evaluation was conducted retrospectively, and a cost minimization study compared asynchronous tele-expertise to face-to-face consultations that would have been performed without ASTE.

### Main outcome measures

In our study we assessed the feasibility of ASTE, what were the origins of the requests for expertise, whether patients need to be moved and the reasons for doing so, and the costs of tele-expertise and conventional consultation.

### Results

In this retrospective analysis 322 advices from 260 patients were interpreted remotely via a platform. The results revealed a 90.68% feasibility of transmitting in a satisfactory and inter-pretable way ultrasound images and videos via the tele-expertise platform (292/322 files). In our series, asynchronous analysis allowed the required physician to make an accurate diagnosis and identify 74 (28.5%, 95% CI [23% –33.9%]) pregnancies associated with malformations and rule out abnormalities in 186 (71.5%, 95% CI [66.1% –77%]) of the cases. The ASTE was not associated with face-to-face consultations for 72.7% (189/260) of the

**Funding:** The author(s) received no specific funding for this work.

**Competing interests:** The authors have read the journal's policy and have the following competing interests: Edwin Quarello is a member of the medical committee of Rofim, a telemedicine startup based in France. There are no patents, products in development or marketed products associated with this research to declare. This does not alter our adherence to PLOS ONE policies on sharing data and materials.

patients, who without moving, were able to have access to a precise diagnosis by ruling out the presence of anomalies in 163/189 of these patients and confirming them in 26/189 patients. The practice of ASTE would result from a societal point of view, an average saving of 61.8% (€ 120.57) per patient compared to a face-to-face consultation.

## Conclusion

The use of asynchronous tele-expertise (ASTE) using fetal ultrasound, is feasible and may contribute to increased diagnostic accuracy while generating a significant reduction in costs for society.

## Introduction

Ultrasound in pregnancy is mainly represented by two components: screening and diagnosis of congenital anomalies. In France, screening is performed by midwives and physicians that usually refer patients to experts or tertiary centers in case of discovery of unusual images and/or malformations. This expertise often geographically dispersed can be requested via tele-expertise, to ensure equitable access to quality care.

A component of telemedicine, tele-expertise is a medical decision-making aid provided to a so-called "requesting" medical professional by one or more medical professionals located at a distance called "required" or expert(s) [1]. This exchange via a secured messaging system can be done in real time "synchronous" or in deferred time "asynchronous". While this practice has always existed, informally, it was only defined and regulated in France in 2010 and reimbursed by National Health Insurance only in February 2019 [2]. Models of telemedicine use have been successfully adopted in many health systems since the 1990s [3–6]. Landwehr et al. noted in 1997 that the development of technologies could improve the concordance of pre and postnatal diagnosis via the visualization of images [3]. Fisk et al. noted in 1996 that the cost of technology was high at the time [5]. These constraints have since been overcome with the development of information and communication technologies. More recent studies have thus shown the technical feasibility of transmitting images from fetal ultrasounds without any real loss of clinically relevant information and of interpreting them precisely [7–10] all by limiting anxiety [11] and unnecessary travel for patients who often live-in regions far from the center of expertise [12–14].

The aim of our work was to assess the use of asynchronous tele-expertise (ASTE) using ultrasound in the setting of prenatal diagnosis according to three axes: the feasibility, the medical relevance of this mode of communication between health professionals, and the cost of such a process from a societal point of view.

## Materials and methods

A retrospective analysis was conducted in 2020 about patients who sought remote advice requests over a 48-month period from January 12, 2016, to January 21, 2020. All these patients were seen in consultations by a midwife sonographer (AL) located at Hyères, 82 km from the expert (EQ) center. He's an experienced practitioner, graduated in national obstetrical ultrasound diploma. He has been performing obstetrical ultrasound for 16 years and this has been his main activity for 13 years.

Usually, when unusual images or even one or more malformations were discovered during a screening ultrasound, patients were sent to a face-to-face consultation for ultrasound control

and prenatal interview with an expert. Faced with this type of situation, the midwife systematically recommends to the patients a second advice made remotely between him and the expert. After having provided the information and obtaining the patient's verbal consent, the midwife sonographer, or requesting, sent the image(s) and/or video loop(s) that he/she deemed appropriate to the expert (required), for the request for advice associated with one or two questions: "Is this usual?" If not, "Should the patient need to be seen face-to-face consultation for additional ultrasound and prenatal advice?". All the requests for advice were realized and recorded on the TriceFy® platform certified as a health data host with CE marking, via the sending of a question associated with images and/or video loops from the various ultrasound examinations.

The tele-expertise requests were classified according to maternal (maternal age at the time of the request), demographic (distance between the municipality of origin of the patient and the expert center in km) as well as fetal (gestational age and the type of anatomical system concerned: central nervous system; face; cardiovascular system; urinary system; digestive system; placenta, amniotic fluid, umbilical cord; etc.) characteristics. In order to assess the tele-expertise advices, the expert classified the still images and/or videos submitted as satisfactory or on the contrary unsatisfactory depending on whether or not an answer could be given to the question asked by the requesting, and thus that an expertise could or could not be carried out. We also collected the number of patients seen in person and the reasons behind these trips. The cost of such a process on an individual and societal scale was estimated.

The clinical data collected by the required doctor has been transmitted anonymously. They were used as the basis for the statistical analysis, using the statistical analysis software "Statistical Package for the Social Sciences" SPSS version 25.0. Descriptive statistics were mainly used. The quantitative variables are presented on average with assessment variables (minimum and maximum) and a dispersion indicator (standard deviation). Qualitative variables are presented in terms of frequency.

A cost minimization analysis comparing the tele-expertise strategy and the conventional strategy was carried out, under the assumption of equivalent efficiency. We compared by decision tree the patients included in our study to a hypothetic group of the same women for whom we simulated that they would have been seen only in a conventional way by going to the expert site. The resources consumed and the results of the strategies compared were identified using the expected impact matrix of the effects of telemedicine [15]. Only direct costs were included, and they were assessed from the perspective of society. For the innovative strategy, the costs taken into account were the tele-expertise act, the platform, the transport, and face-to-face consultation for patients who still required those. In France, four acts of tele-expertise per year and per patient can be billed to the National Insurance at a rate of 10 euros for the person requesting the opinion and 20 euros for the physician requested (expert). For the conventional strategy, the costs taken into account were the face-to-face consultation and the transport, which were applied to all the advices requested by all patients in the study. Cost data were reported as mean values and compared using a t-test. Bootstrapping (1000 replicates) was used to estimate the uncertainty in the average total cost distribution for each strategy. Economic modeling was performed on TreeAge Pro 2020 software. The probability figures for the occurrence of each event were derived from the clinical data of the present study. A sensitivity analysis was performed to test the robustness of the model. The details of the methodology used and the results will be presented in a future publication [16].

## Ethics

The study protocol was approved by Inserm's Ethical Evaluation Committee (opinion n˚19–622).

# Results

## Description of the study population

The midwife requested a tele-expertise for 260 patients, at the origin of 322 advices, 4 of whom were related to twin pregnancies. All patients accepted the ASTE's request. During this study period, the midwife performed in average 2000 obstetrical ultrasound per year.

The mean age of the patients was 31.8 years (± 4.6). The mean gestational age at the time of ASTE was 27.7 (± 6.2) weeks. Of the 322 advices, 18 (5.6%), 112 (34.8%), and 192 (59.6%) were for pregnancies in the first, second, and third trimesters, respectively. Among the 260 patients, 213 (81.9%), 37 (14.2%), 6 (2.3%), 3 (1.2%) and 1 (0.4%) patient benefited from one, two, three, four and five advices respectively (Table 1).

231/260 (88.1%), 25/260 (9.64%), and 4/260 (1.5%) of patients sought advice requests that involved one, two, or three anatomical systems respectively (Table 1).

## ASTE data

**Requests for advice.**   Out of the 322 files sent by the midwife sonographer via the tele-expertise platform 292 (90.7%) were qualified by the expert as satisfactory, thus allowing their

**Table 1. Characteristics of the study population.**

| characteristics | Average (Standard deviation) | | |
|---|---|---|---|
| Maternal age (n = 260) | 31.8 (4.6) | | |
| Gestational age (n = 322) | 27.7 (6.2) | | |
| Advices per patient (n = 260) | 1.24 (0.59) | | |
| Devices concerned per patient (n = 260) | 1.13 (0.38) | | |
| Advices per patient | Frequency (%) | | |
| 1 advice | 213 (81.9) | | |
| 2 advices | 37 (14.2) | | |
| 3 advices | 6 (2.3) | | |
| 4 advices | 3 (1.2) | | |
| 5 advices | 1 (0.4) | | |
| Devices involved for each pregnancy | | | |
| 1 device | 231 (88.9) | | |
| 2 devices | 25 (9.6) | | |
| 3 devices | 4 (1.5) | | |
| Number of notices per trimester of pregnancy / age group (years) | T1 | T2 | T3 |
| < 20 | - | - | 1 |
| 20–30 | 6 | 33 | 61 |
| 30–40 | 12 | 72 | 120 |
| > 40 | - | 7 | 10 |
| Total: 322 reviews (260 patients) | 18 | 112 | 192 |
| Gestational age (SA) / Number of advices by devices concerned | 1 device | 2 devices | 3 devices |
| 9–16 | 18 | 2 | - |
| 16–23 | 59 | 3 | 2 |
| 23–30 | 61 | 5 | 1 |
| 30–37.2 | 157 | 13 | 1 |
| Total: 322 reviews (260 patients) | **295** | **23** | **4** |

The average number of requests sent by the midwife to the expert per month and per quarter during the study period was 7 (min:1, max:18) and 19 (min:36, max:3) respectively.

**Table 2. Data relating to expertise.**

|  | Frequency (%) |
|---|---|
| Quality of the files relating to the 322 reviews |  |
| Satisfactory | 292 (90.7%) |
| Unsatisfactory | 30 (9.3%) |
| Patients seen or not (n = 260) |  |
| Unseen patients | 189 (72.7%) |
| Patients seen | 71 (27.3%) |
| Reasons for travel for a face-to-face consultation (n = 71) |  |
| Additional expertise required—prenatal counseling | 39 (15.0%) |
| Unsatisfactory resolution of transmitted files | 11 (4.2%) |
| Prenatal counseling | 9 (3.5%) |
| Concern REQUIRING despite the conclusion of the required physician | 6 (2.3%) |
| PATIENT'S concern despite the conclusion of the requested physician | 6 (2.3%) |
| Diagnoses made by patients seen or not (n = 260) |  |
| Anomalies detected | 74 (28.5%) |
| Unseen patients | 26 (35.1%) |
| Patients seen | 48 (64.9%) |
| No anomalies detected | 186 (71.5%) |
| Unseen patients | 163 (87.6%) |
| Patients seen | 23 (12.4%) |

interpretation and answering the question asked by the applicant. Given there is, so far, no guideline related to remote expertise, we considered the expert's point of view. For the remaining 30 (9.3%) files, it was not possible to answer the question asked (Table 2). This required face-to-face technical expertise from the expert in order to obtain additional images necessary for the establishment of the diagnosis and or in connection with poor quality of the still images and or videos sent. Referral for prenatal counseling is related to the severity of the fetal anomaly or the degree of concern the couple has about the diagnosis (Table 2).

A total of 72 (22.4%) of the 322 consultations required patients to travel. The distribution of the 322 consultations by type of consultation, ASTE alone or ASTE combined with a face-to-face consultation, is shown in (Fig 1). Data is broken down by calendar quarter.

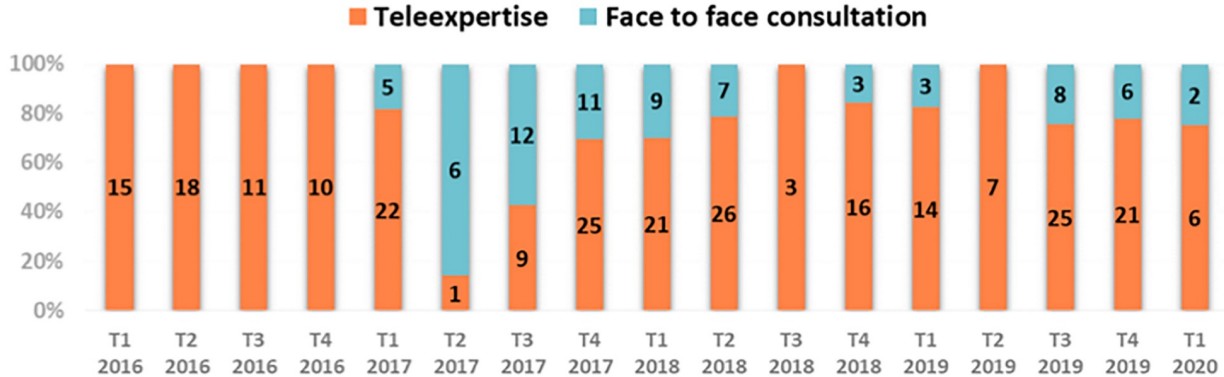

**Fig 1. Breakdown by calendar quarter of the 322 advices requested by the 260 patients according to whether they were seen face-to-face or tele-expertized.**

**Patients.** If we consider now the 260 patients and no longer the 322 requests for advice, the advices from 232 (89.2%) and 28 (10.8%) of the patients were considered to be satisfactory and unsatisfactory, respectively. In the 260 patients, we observed an absence and presence of abnormalities in 186 (71.5%) and in 74 (28.5%) patients respectively; 46 of these patients were seen in the clinic and the reasons are detailed in (Fig 2).

**Face-to-face cases.** 71/260 (27.3%) patients, at the origin of 97 advices, were seen in face-to-face consultation. However, they were seen for 72/97 (74.2%) of these advices (one patient seen twice). 25/97 (25.8%) of the remaining corresponded to second tele-expert advices later during the pregnancy.

The reasons for the face-to-face consultation are represented by the following situations: 11/71 (15.5%) of the patients seen are related to an inability to answer the question asked due to insufficient and or a poor quality of the images and or videos transmitted. 39/71 (54.9%) were seen for a complementary ultrasound examination to ensure the isolated nature of the

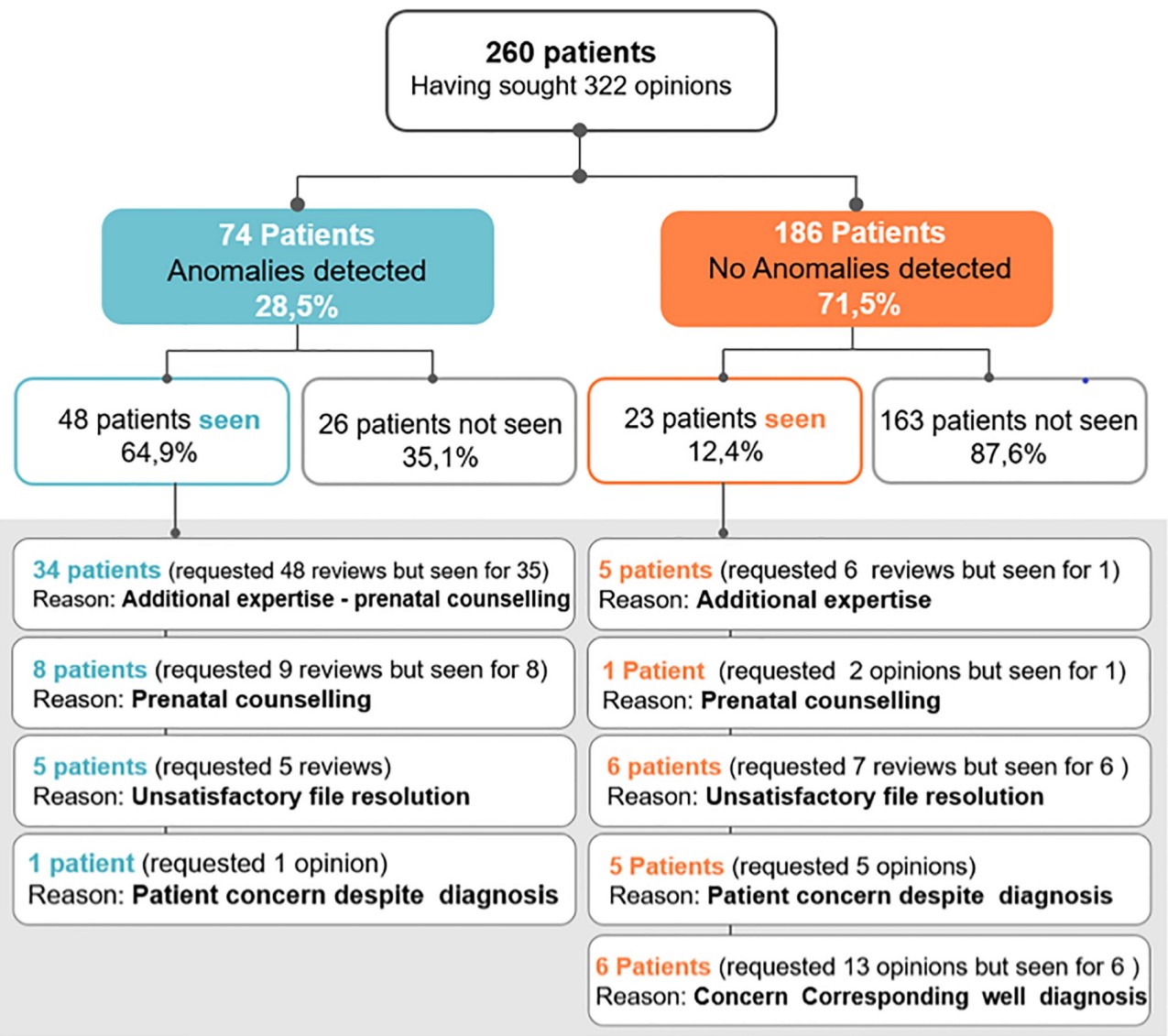

**Fig 2. Breakdown of 260 patients who benefited from the tele-expertise strategy, with specific reasons for face-to-face consultations.**

**Table 3. Distribution of malformations detected by type of device concerned.**

| Isolated malformations | | Associated malformations | |
|---|---|---|---|
| Heart | 15 | Heart +Appendices | 1 |
| CNS | 11 | Heart +CNS | 1 |
| NT | 6 | Heart +Digestive | 1 |
| Vascular | 6 | Heart + NT | 1 |
| Face | 5 | Heart + Urinary | 1 |
| Digestive | 5 | CNS + Appendices | 1 |
| Urinary | 4 | CNS + Digestive | 1 |
| Skeleton | 3 | CNS + heart + Digestive | 1 |
| Ends | 2 | CNS +Biometrics | 1 |
| Appendices | 2 | CNS + ExG | 1 |
| Biometrics | 1 | Urinary + IUGR | 1 |
| Thorax | 1 | Skeleton + IUGR | 1 |
| Polymalformative syndrome | 1 | | |

CNS: Central nervous system, NT: Nuchal translucency, IUGR: Intrauterine growth restriction, ExG: Malformation of the external genitalia.

anomaly accompanied by an antenatal interview. 9/71 (12.7%) required an antenatal counselling interview despite the fact that the images allowed a diagnosis to be made. 6/71 (8.5%) were seen due to the concern of the patient and 6/71 (8.5%) were seen due to the concern of the requesting, despite the fact that an expert's opinion may have already been established (Table 2). It is important to underline that 21/71 (29.6%) seen in face-to-face already had a diagnosis established by ASTE.

In 189/260 (72.7%) patients, the assessment of images and/or videos was deemed satisfactory by the expert by ASTE and physical consultations could have been avoided.

The anomalies detected are reported separately and in combination in Table 3.

## ASTE economic analysis

Over the 48-month study period and for a population of 260 patients, we estimated the total cost from a societal point of view of the innovative strategy to € 19,356.32 compared to € 50,707.40 for the conventional strategy. Thus, the average total cost per patient of the ASTE strategy was € 74.45 (95% CI: € 66.36–€ 82.54) against € 195.02 (95% CI: € 183.90–€ 206.14) for the conventional one. The practice of ASTE would result from a societal point of view, an average saving of 63,27% (€ 123.40) per patient.

In the conventional strategy (single face-to-face consultation) we considered two cost factors: the consultation and the transport that represented 61.6% (€ 120.08) and 38.4% (€ 74.94) respectively of the average total cost. With regard to ASTE, whose operating mode mainly combines advice given at a distance but also a lesser part of face-to-face consultation, we considered four cost factors at the origin of the costs: the parts attributable to the act of tele-expertise, the face-to-face consultation when it is carried out, the transport associated with this consultation, and the management costs of the tele-expertise platform representing 41.8% (€ 31.15), 34.6% (€ 25.73), 22.0% (€ 16.41), and 1.6% (€ 1.16) respectively of the average total cost. The share of each cost by strategy is illustrated in (Fig 3).

In addition to the reduction of costs, the ASTE strategy notably made it possible to save time for these 260 patients who would have covered an average distance of 158.98 km with a

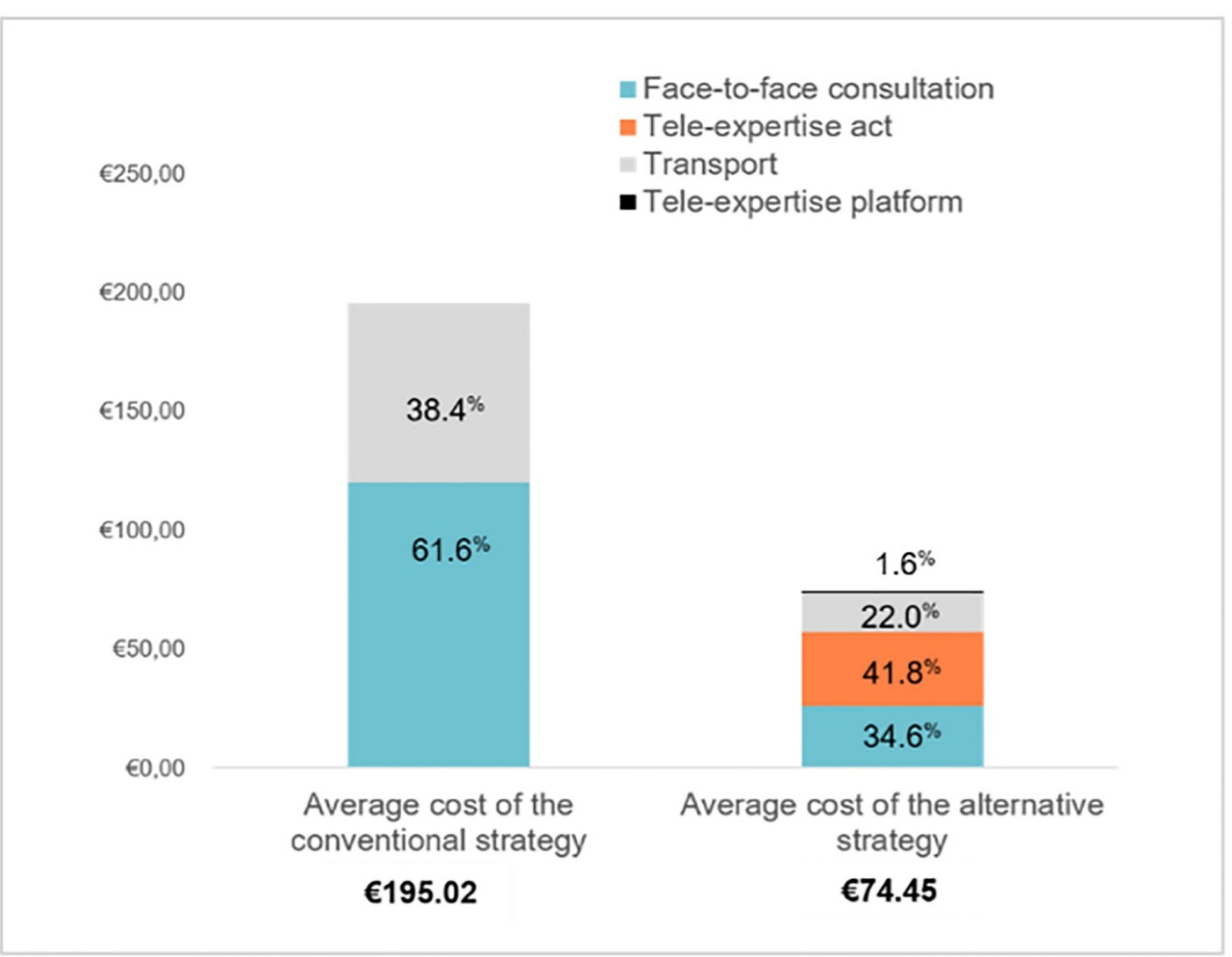

**Fig 3. Distribution of average costs by strategy.**

minimum and maximum distance of 92.7 km and 496 km respectively. The average journey time was estimated at 2h19mn (min: 1h34mn, max 6h34mn).

## Discussion

This study highlights the potential of ASTE using ultrasound to facilitate the exchange of medical data between health professionals, allowing for timely and non-time-consuming accurate remote expertise without requiring the presence or travel of patients. Indeed, obstetric ultrasounds and the resulting data seem suitable for ASTE as a transmission model necessary for their remote interpretation, this is corroborated by the findings, on the one hand, of a high rate of interpretability of 90.7% (95% CI = [87.5%; 93.9%]) and on the other hand, ASTE detected 28.5% (95% CI = [23.0%; 33.9%]) abnormalities in our study population. This practice also has an economic interest, by reducing the direct costs generated by a face-to-face consultation.

The advantage that this technology offers is twofold because besides the confirmation of an abnormality which is essential before the postnatal management options are discussed with the parents, the exclusion of a false positive case is also an equally important clinical criterion.

Indeed, ASTE can significantly contribute to reduce the psychological stress that could ultimately be induced by a false diagnosis by considerably reducing the time required to obtain an expert opinion. The present study highlights in particular a significant reduction in the number of patient trips, with the possible recourse to a face-to-face consultation which will always be necessary for a minority of cases [5]. These cases mainly require technical expertise and for reassurance, which can only be obtained through meeting with the expert. Patients who were seen for an explanatory consultation could now be conducted by videoconference. This modality was not retained during the study period, because while we had voluntarily agreed to handle requests for advice via ASTE, we had chosen to continue to see couples face-to-face when a significant problem was discovered.

The majority of abnormalities detected by the expert physician were cardiac. This type of serious malformation is among the most common and their detection is sometimes complex [17]. To overcome the limited availability of this expertise, F. Viñals et al. evaluated, in this regard, the correlation between the interpretations from spatio temporal image correlation (STIC) volumes obtained by two operators (obstetricians) inexperienced in fetal echocardiography, transmitted via the Internet. The authors concluded that an Internet tele-expertise relationship is technically feasible and is a very useful tool for the diagnosis and learning of congenital heart disease [12]. This teaching effect and these same types of clinical outcomes have also been reported by McCrossan et al. [13].

Thus, tele-expertise may help improve the learning curve of the requesting physician, especially in the presence of successful interprofessional communication (via email or telephone). However, soon the acquisition of synchronous images or videos could allowed more control of the quality in real time and reduced the frequency of transmitted files judged to be uninterpretable but are, so far, difficult to schedule. Of course, the Body Mass Index (BMI) is an important cause, not evaluated here, that can be the cause of sending uninterpretable images and videos. In the construction of the relationship between the requesting and the required, it is important to underline the construction of a learning curve in the way of asking a question and especially in the way of associating informative images and videos.

The establishment of training programs and certification in imaging subject to tele-expertise should be given priority in the near future in order to improve the skills development of inexperienced operators. Vinayak et al. demonstrated that midwives who had no previous training in ultrasound could be trained in the use of ultrasound and perform obstetric ultrasounds, whose images and interim reports were transmitted over a 3G mobile phone network. The results showed excellent correlations between prenatal and postnatal diagnoses [18].

However, consulting a colleague when you feel that you do not have the necessary expertise to make a diagnosis is certainly a delicate exercise, but essential for the accuracy of the diagnosis [19]. In addition to the current limits underpinned by the acceptance of changing the method of requesting an opinion, there is an issue relating to the remuneration of tele-expertise acts. When it is considered too low by health professionals, it constitutes a major obstacle to their adherence [20]. All these existing reasons contributed to the unattractiveness of tele-expertise in the past and even to this day, with regard to the tele-expertise practices carried out [21] and the medico-legal risk without any specific case law to date in France. However, the current pandemic has changed the paradigm of all medical appointment modalities, and we are witnessing a real craze for this new type of practice.

Although evaluated here for prenatal diagnosis by obstetrical ultrasound, our model can be applied to other specialties based on still images or video loops such as dermatology, ophthalmology and radiology. The development of a tele-expertise model, as described in our study, easy to implement in clinical practice, inexpensive (require an inexpensive technology, not requiring specific equipment), and with effective clinical outcomes, would allow a

rationalization of resources and equitable access to expertise which is often geographically dispersed. Nevertheless, it requires a good understanding of the limits associated with this mode of communication, and above all a relationship of trust between the "requesting-requested" partners. Actually, good coordination supported by interactive communication, for example, by telephone appointments or e-mail exchanges, would contribute more to facilitating the exchange and thus bring about an improved collaboration that is synonymous with mutual trust.

However, in order not to be faced with an inequality of access to care, this time at a distance, the establishment of a regional and then national network of requests for advices with exchanges between the requesting and required people through various applications could, in the long term, be put in place so that advices emanating from a region are automatically directed to experts emanating from private or public structures originating from these same regions.

The main limitations of our study were related to the retrospective nature of the data collection, and to the difficulties in recovering the outcomes of all pregnancies of patients who benefited from one or more advices.

Furthermore, we were unable to calculate the sensitivity and specificity of the diagnoses made because the cases considered negative by the requesting person, and which are necessary for these calculations, were not referred to the required doctor. However, we calculated a kappa agreement rate of 1 indicating perfect agreement between prenatal diagnoses and postnatal outcomes for the subgroup of patients (141 patients associated with 174 advices) for whom we were able to have postnatal results during the period 2018–2020 [22].

However, the extended study period offered the advantage of increasing the number of patients studied to have a fairly representative sample size with real-life clinical data that best reflects standard practice. A future prospective study will allow to assess patients, midwives, as well as physicians' perception of tele-expertise in order to measure their adherence and to question them on the renunciation of care avoided thanks to this process.

## Conclusion

The use of asynchronous tele-expertise (ASTE) using ultrasound, in the field of prenatal diagnosis, is feasible and may contribute to increased diagnostic accuracy while generating a significant reduction in costs for society.

## Supporting information

**S1 File.**
(XLSX)

## Author Contributions

**Conceptualization:** Edwin Quarello.

**Data curation:** M'hamed Beldjerd, Antoine Lafouge, Edwin Quarello.

**Formal analysis:** M'hamed Beldjerd, Anne-Gaëlle Le Corroller-Soriano, Edwin Quarello.

**Methodology:** M'hamed Beldjerd, Anne-Gaëlle Le Corroller-Soriano, Edwin Quarello.

**Software:** M'hamed Beldjerd.

**Supervision:** Edwin Quarello.

**Validation:** Anne-Gaëlle Le Corroller-Soriano.

**Writing – original draft:** M'hamed Beldjerd.

**Writing – review & editing:** Roch Giorgi, Anne-Gaëlle Le Corroller-Soriano, Edwin Quarello.

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
