## [Decision Letter · Decision Letter 0]

8 Mar 2022

PONE-D-21-40075Asynchronous tele-expertise (ASTE) for prenatal diagnosis is feasible and cost saving: results of a French case studyPLOS ONE

Dear Dr. Quarello,

Thank you for submitting your manuscript to PLOS ONE. After careful consideration, we feel that it has merit but does not fully meet PLOS ONE’s publication criteria as it currently stands. Therefore, we invite you to submit a revised version of the manuscript that addresses the points raised during the review process.

We look forward to receiving your revised manuscript.

Kind regards,

David Desseauve, MD, MPH, PhD

Academic Editor

PLOS ONE

Journal Requirements:

3. If materials, methods, and protocols are well established, authors may cite articles where those protocols are described in detail, but your submission should include sufficient information to be understood independent of these references (https://journals.plos.org/plosone/s/submission-guidelines#loc-materials-and-methods).

6. Please include a copy of Table 2 and 3 which you refer to in your text on page 11, 12 and 13.

Reviewers' comments:

Reviewer's Responses to Questions

**Comments to the Author**

1. Is the manuscript technically sound, and do the data support the conclusions?

Reviewer #1: Yes

Reviewer #2: Yes

2. Has the statistical analysis been performed appropriately and rigorously? 

Reviewer #1: I Don't Know

Reviewer #2: I Don't Know

3. Have the authors made all data underlying the findings in their manuscript fully available?

Reviewer #1: Yes

Reviewer #2: Yes

4. Is the manuscript presented in an intelligible fashion and written in standard English?

Reviewer #1: Yes

Reviewer #2: No

5. Review Comments to the Author

Reviewer #1: Dear Editor

Thank you for the opportunity of reviewing this very interesting study concerning Asynchronous tele-expertise (ASTE) for prenatal diagnosis. This study highlights the potential of ASTE using ultrasound to facilitate the exchange of medical data between health professionals in obstetrics and to limit useless travels. According to the authors, this practice also has an economic interest, by reducing the direct costs generated by a face-to-face consultation. This topic is very important, especially in areas with limited recourses to expert centers. Publications of these preliminary results could be relevant to encourage this practice and to help medical institutions in supporting diffusion of this expertise. Reassuring results concern quality of the transferred data because 90.7% were qualified by the expert as satisfactory, thus allowing their interpretation and answering the question asked by the applicant and 29.6% of women seen in face to face already had a diagnosis established by ASTE.

I propose the following comments to improve the quality of the paper.

- Inclusion criteria to propose ASTE should be clarified. It is not clear when a ASTE is proposed by requesting” medical professional. During the study period 260 women were required ASTE, but how many women were screened during the studied preriod. Is an ASTE systematically required after prenatal detection of an abnormal image?

- In the method section, it would be important to give informations about the profile of the requiring professional. We understand that the professional is a midwife, but what is his academic background? number of year of practices? These elements are important to interpret external validity of this practice and its transferability in others clinical contexts

- One limitation of the study concerns absence of a systematical collection of neonatal outcomes to corelate pre and postnatal data

- Concerning maternal characteristics, BMI is an important data to interpret quality of the transferred files

- 90.7% of transferred files were qualified as satisfactory by the expert, according to what? National guidelines, ISUOG guidelines?

- Concerning medico-economic analysis, authors considered two cost factors: the consultation and the transport that represented 62% (€ 121.91) and 38% (€ 74.11) respectively of the average total cost. What does the amount of 121.91 euros correspond to? Average ultrasound cost in France?

Reviewer #2: The article of Beldjerd M. and colleagues is very interesting and of very current interest. The development of telemedicine in prenatal diagnosis is of significant benefit for remote regions but also from a societal point of view apart the mentioned benefits described in the article, by reducing the repeated absences from the workplace of the parents and by allowing very probably, when the system is well organized, to reduce the waiting time and anxiety linked to the sonographer's doubts.

Corrections:

1. I think that the position of E. Quarello in the medical committee of ROFIM company should be mentioned under competing interests.

2. In the abstract

2.1 under Main Outcome TEAS should be replaced by ASTE.

2.2 under results "asynchronous analysis prevented the displacement of 72.7% (189/260)" 189 is difficult to understand. I would say X/malformations and X/ no malformations had no face to face.

2.3 Average saving should be in percentage (see under)

3. In Results

3.1 The text of this whole chapter is too redundant with figures and tables. Some of the results could be presented in the text and the others only in the figures and tables

3.2 In the "face to face cases", the sentence "232 292/322 (90.7%) requests were qualified as satisfactory, thus allowing their interpretation and answering the question asked by the applicant" is already mentioned in "Request for advice".

3.3 Economic analysis is the central and also motivating point for implementing tele-expertise. Nevertheless the analysis in Euro is only understandable in France for those who know the local costs but for a more international understanding, I would express the decrease in costs as a percentage and it would also be interesting to know the reimbursement of the ASTE in France and the percentage of economy compared to the cost of face to face examination carried out directly by the expert.

I fully agree with the conclusions advanced by the authors.

Not being a native English speaker, I cannot comment on the writing of the article but it seems to me that certain passages should benefit from proofreading by an English speaker.

6. PLOS authors have the option to publish the peer review history of their article (what does this mean?). If published, this will include your full peer review and any attached files.

Reviewer #1: No

Reviewer #2: No

---

## [Author Response · Author response to Decision Letter 0]

27 Mar 2022

Dear Editor,

Thank you for giving us the opportunity to submit our work. Please find hereby the answers to the reviewers’ requests.

Sincerely yours

Edwin QUARELLO

Reviewer #1: 

Dear Editor

Thank you for the opportunity of reviewing this very interesting study concerning Asynchronous tele-expertise (ASTE) for prenatal diagnosis. This study highlights the potential of ASTE using ultrasound to facilitate the exchange of medical data between health professionals in obstetrics and to limit useless travels. According to the authors, this practice also has an economic interest, by reducing the direct costs generated by a face-to-face consultation. This topic is very important, especially in areas with limited recourses to expert centers. Publications of these preliminary results could be relevant to encourage this practice and to help medical institutions in supporting diffusion of this expertise. Reassuring results concern quality of the transferred data because 90.7% were qualified by the expert as satisfactory, thus allowing their interpretation and answering the question asked by the applicant and 29.6% of women seen in face to face already had a diagnosis established by ASTE.

I propose the following comments to improve the quality of the paper.

- Inclusion criteria to propose ASTE should be clarified. It is not clear when a ASTE is proposed by requesting” medical professional. During the study period 260 women were required ASTE, but how many women were screened during the studied period. We added this information in the Results section. Indeed, on average the midwife performs 2000 ultrasounds per year (Line 167 – 168). Is an ASTE systematically required after prenatal detection of an abnormal image? As mentioned in this section, ASTE was systematically proposed when unusual images are discovered (Line 115).

- In the method section, it would be important to give information about the profile of the requiring professional. We understand that the professional is a midwife, but what is his academic background? number of year of practices? These elements are important to interpret external validity of this practice and its transferability in others clinical contexts. We added the information related to the midwife experience in the Material and Methods section as well (Line 109 – 111).

- One limitation of the study concerns absence of a systematical collection of neonatal outcomes to corelate pre and postnatal data. We highlighted this notion by mentioning in the Discussion the perfect correlation between pre and postnatal data in a previous preliminary study (Asynchronous Tele-Expertise (ASTE) in obstetrical ultrasound: Is it equivalent to face-to-face consultation?. Beldjerd MH, Lafouge A, Le Corroller Soriano AG, Quarello E. Gynecol Obstet Fertil Senol. 2021 Nov;49(11):850-857.) (Line 343 – 346)

- Concerning maternal characteristics, BMI is an important data to interpret quality of the transferred files. Yes, you’re right and we mentioned this in the Discussion (Line 297). Moreover, in the construction of the relationship between the requesting and the required, it is important to underline the construction of a learning curve in the way of asking a question and especially in the way of associating informative images and videos. We added this in the Discussion as well (Line 299 – 302).

- 90.7% of transferred files were qualified as satisfactory by the expert, according to what? National guidelines, ISUOG guidelines? 90.7% of files were classified by the expert as satisfactory related to their ability to answer to the following questions: “Is this usual? ” If not, “ Should the patient need to be seen face-to-face consultation for additional ultrasound and prenatal advice?”. Given there is, so far, no guideline related to remote expertise, we considered the expert’s point of view. This was mentioned in the Results section (Line 194 – 196).

- Concerning medico-economic analysis, authors considered two cost factors: the consultation and the transport that represented 62% (€ 121.91) and 38% (€ 74.11) respectively of the average total cost. What does the amount of 121.91 euros correspond to? Average ultrasound cost in France? As mentioned in the Results section, In the conventional strategy (single face-to-face consultation) we considered two cost factors: the consultation and the transport that represented 61.6% (€ 120.08) and 38.4% (€ 74.94) respectively of the average total cost (Line 247 – 249).

Reviewer #2: 

The article of Beldjerd M. and colleagues is very interesting and of very current interest. The development of telemedicine in prenatal diagnosis is of significant benefit for remote regions but also from a societal point of view apart the mentioned benefits described in the article, by reducing the repeated absences from the workplace of the parents and by allowing very probably, when the system is well organized, to reduce the waiting time and anxiety linked to the sonographer's doubts.

Corrections:

1. I think that the position of E. Quarello in the medical committee of ROFIM company should be mentioned under competing interests. We mentioned this in Disclosure section.

2. In the abstract

2.1 under Main Outcome TEAS should be replaced by ASTE. Thank you, we corrected this mistake.

2.2 under results "asynchronous analysis prevented the displacement of 72.7% (189/260)" 189 is difficult to understand. I would say X/malformations and X/ no malformations had no face to face. We completed it (Line 62-65)

2.3 Average saving should be in percentage (see under). We modified it as requested (Line 66).

3. In Results

3.1 The text of this whole chapter is too redundant with figures and tables. Some of the results could be presented in the text and the others only in the figures and tables. We have deliberately chosen to put some data in the Results section as well as in the Tables. Our current distribution seems to be good.

3.2 In the "face to face cases", the sentence "232 292/322 (90.7%) requests were qualified as satisfactory, thus allowing their interpretation and answering the question asked by the applicant" is already mentioned in "Request for advice". Thank you, we deleted this sentence here.

3.3 Economic analysis is the central and also motivating point for implementing tele-expertise. Nevertheless the analysis in Euro is only understandable in France for those who know the local costs but for a more international understanding, I would express the decrease in costs as a percentage (We modified it as requested Line 245 – 246) and it would also be interesting to know the reimbursement of the ASTE in France (Since our initial submission, the remuneration for tele-expertise acts has evolved, going from two levels of remuneration (depending on the complexity of the medical case) to a single level. Indeed, the first level provided that the requesting doctor received 5 euros and the required doctor 12 euros. For the second, they receive respectively 10 and 20 euros. Payments were made on a fee-for-service basis for the requested doctor and in the form of a limited annual lump sum for the requesting doctor. Now, the requesting doctor gets €10 and the required doctor €20, within the limit of 4 acts per year, per required doctor, for the same patient. Even if this change was taken into account in our medico-economic assessment through the sensitivity analysis, we preferred to update our data taking into account the new pricing. The tele-expertise strategy remains less costly. We added this information Line 148 – 151 in the Material and Methods section.and the percentage of economy compared to the cost of face to face examination carried out directly by the expert. We added this information Line 212 – 213 in the Results section.

I fully agree with the conclusions advanced by the authors.

Not being a native English speaker, I cannot comment on the writing of the article but it seems to me that certain passages should benefit from proofreading by an English speaker. We submitted this article to a native English speaker and he made minimal edits.

---

## [Decision Letter · Decision Letter 1]

23 May 2022

Asynchronous tele-expertise (ASTE) for prenatal diagnosis is feasible and cost saving: results of a French case study

PONE-D-21-40075R1

Dear Dr. Quarello,

We’re pleased to inform you that your manuscript has been judged scientifically suitable for publication and will be formally accepted for publication once it meets all outstanding technical requirements.

Kind regards,

David Desseauve, MD, MPH, PhD

Academic Editor

PLOS ONE

Reviewers' comments:

Reviewer's Responses to Questions

**Comments to the Author**

1. If the authors have adequately addressed your comments raised in a previous round of review and you feel that this manuscript is now acceptable for publication, you may indicate that here to bypass the “Comments to the Author” section, enter your conflict of interest statement in the “Confidential to Editor” section, and submit your "Accept" recommendation.

Reviewer #1: All comments have been addressed

Reviewer #2: All comments have been addressed

2. Is the manuscript technically sound, and do the data support the conclusions?

Reviewer #1: Yes

Reviewer #2: Yes

3. Has the statistical analysis been performed appropriately and rigorously? 

Reviewer #1: N/A

Reviewer #2: Yes

4. Have the authors made all data underlying the findings in their manuscript fully available?

Reviewer #1: Yes

Reviewer #2: Yes

5. Is the manuscript presented in an intelligible fashion and written in standard English?

Reviewer #1: Yes

Reviewer #2: Yes

6. Review Comments to the Author

Reviewer #1: The authors took into account all my previous recommendations and all comments have been addressed to improve quality of the paper

Reviewer #2: No other comments. Authors made their article sound better and they answered to all comments in an excellent way.

7. PLOS authors have the option to publish the peer review history of their article (what does this mean?). If published, this will include your full peer review and any attached files.

Reviewer #1: No

Reviewer #2: No

---

## [Editor Report · Acceptance letter]

21 Jul 2022

PONE-D-21-40075R1 

Asynchronous tele-expertise (ASTE) for prenatal diagnosis is feasible and cost saving: results of a French case study 

Dear Dr. Quarello:

I'm pleased to inform you that your manuscript has been deemed suitable for publication in PLOS ONE. Congratulations! Your manuscript is now with our production department. 

Kind regards, 

on behalf of

Dr. David Desseauve 

Academic Editor

PLOS ONE